# Household Income Is Associated with Chronic Pain and High-Impact Chronic Pain among Cancer Survivors: A Cross-Sectional Study Using NHIS Data

**DOI:** 10.3390/cancers16162847

**Published:** 2024-08-15

**Authors:** Nimish Valvi, Javier A. Tamargo, Dejana Braithwaite, Roger B. Fillingim, Shama D. Karanth

**Affiliations:** 1Department of Nutrition and Health Science, Ball State University, Muncie, IN 47306, USA; nimish.valvi@bsu.edu; 2Pain Research and Intervention Center of Excellence, University of Florida, Gainesville, FL 32611, USA; j.tamargo@ufl.edu (J.A.T.); rfilling@ufl.edu (R.B.F.); 3Institute on Aging, University of Florida, Gainesville, FL 32611, USA; 4Department of Community Dentistry and Behavioral Science, University of Florida, Gainesville, FL 32611, USA; 5Department of Surgery, College of Medicine, University of Florida, Gainesville, FL 32611, USA; 6University of Florida Health Cancer Center, University of Florida, Gainesville, FL 32611, USA

**Keywords:** social determinants of health, health disparities, cancer pain, pain management

## Abstract

**Simple Summary:**

Pain is a common symptom that affects individuals with cancer. This study, using data from the National Health Interview Survey, explored how household income relates to chronic pain among cancer survivors. It found that lower-income survivors, those below 200% of the federal poverty level (FPL), were more likely to report chronic pain lasting at least 3 months and pain that significantly limited their daily activities. Compared to higher-income survivors (at least 400% FPL), those with lower incomes had higher odds of experiencing chronic pain and its impact on daily life. Additionally, opioid use for pain management was more common among lower-income survivors, while higher-income survivors tended to use alternative methods like yoga, chiropractic care, and physical therapy. This study emphasizes the need for targeted efforts to address healthcare disparities and improve pain management for all cancer survivors, regardless of their income level.

**Abstract:**

Pain is a prevalent issue among cancer patients, yet its link with socioeconomic status has not been thoroughly examined. This study investigated chronic pain (lasting ≥3 months) and high-impact pain (chronic pain limiting activities) among cancer survivors based on household income relative to the federal poverty level (FPL), using data from the National Health Interview Survey (2019–2020). Of the 4585 participants with a history of solid cancers, 1649 (36.3%) reported chronic pain and 554 (12.6%) reported high-impact chronic pain. After adjustment, participants with incomes < 200% FPL had significantly higher odds of chronic pain (adjusted odds ratio [aOR]: 1.60, 95% CI: 1.25–2.05) and high-impact chronic pain (aOR: 1.73, 95% CI: 1.09–2.74) compared to those with incomes ≥ 400% FPL. Opioid use for chronic pain was most prevalent among those with incomes < 200% FPL (28.3%) compared to those with 200–399% (21.3%) and ≥400% (19.0%). Higher-income participants reported greater use of alternative pain management methods such as yoga (50.5%), chiropractic care (44.8%), and physical therapy (44.3%). This study highlights the association between household income and chronic pain outcomes among cancer survivors, emphasizing the necessity for targeted interventions to mitigate healthcare access disparities and improve pain management for all individuals affected by cancer.

## 1. Introduction

In recent decades, the decline in cancer-related mortality rates has resulted in a substantial increase in the number of long-term cancer survivors [1]. This increase can be attributed to multi-level factors, including an aging population, advances in cancer screening, and the development of newer, more effective treatments that have reduced cancer-specific mortality [2]. With this expanding population of cancer survivors comes a pressing need to comprehensively understand the long-term side effects they experience. Among these effects, chronic pain has emerged as a particularly prevalent symptom, profoundly impacting their quality of life [3,4,5]. While chronic pain affects an estimated 21% of United States (U.S.) adults, it afflicts 35% of cancer survivors [6]. Furthermore, cancer survivors also experience pain-related disability at a disproportionately higher rate, with 13% reporting high-impact chronic pain (i.e., pain that limits life and work activities) compared to just 7% of the general U.S. adult population [6]. As such, understanding factors that contribute to the disproportionate burden of chronic pain among cancer survivors is of paramount importance.

Socioeconomic status (SES) stands as a crucial determinant for both cancer incidence and chronic pain [7], yet the intricate interplay of SES with chronic pain among cancer survivors remains insufficiently explored. Cancer survivors constitute a unique cohort grappling with multifaceted challenges and responsibilities, which often precipitate financial strains [8,9,10]. These may stem from mounting medical costs, cancer-induced disabilities, familial caregiving duties, and the imperative of maintaining professional commitments. The added burden of financial toxicity significantly compounds the complexities of managing chronic cancer pain, particularly among those with inadequate insurance coverage and economically disadvantaged backgrounds. Low SES not only heightens vulnerability to chronic pain but may also shape approaches to pain management [11].

The objective of this study was to investigate the relationship between SES and chronic pain among a nationally representative sample of U.S. adult cancer survivors. Specifically, we evaluated the prevalence of chronic pain and high-impact chronic pain and the utilization of pain management methods among cancer survivors, stratified by household income relative to the federal poverty level (FPL).

## 2. Materials and Methods

### 2.1. Study Design and Population

This cross-sectional study analyzed combined data from the 2019–2020 National Health Interview Survey (NHIS) to explore the association between household income and chronic pain among solid cancer survivors. The NHIS data comprise annual, cross-sectional household interview national surveys that utilize complex, multi-stage sampling methodologies to generate estimates regarding the noninstitutionalized population of the United States [12]. The NHIS interviews are conducted using computer-assisted personal interviewing. Face-to-face interviews are conducted in respondents’ homes, but follow-ups to complete interviews may be conducted over the telephone. A comprehensive explanation of the NHIS sampling technique and the procedures for data collection can be accessed via the following link: https://www.cdc.gov/nchs/nhis/index.htm (accessed on 31 January 2024).

This study utilized information derived from the Sample Adult files, including participants who were aged 18 years and older [12]. All NHIS data featured in this research are based on self-reported diagnosis of solid cancer and participant responses regarding chronic pain and pain management methods. Participants with the following characteristics were excluded from the analysis: (1) participants with no history of solid cancers and (2) those with missing or invalid data (e.g., refused, not ascertained, do not know) on chronic pain. The NHIS is a valuable resource for chronic pain surveillance in the U.S., with questions specifically designed to align with the International Association for the Study of Pain’s definitions of chronic and high-impact chronic pain [13]. A flow chart illustrating the inclusion and exclusion criteria for participants can be seen in Figure 1. The dataset comprises sampling weights designed to adjust for the cluster design, non-response, and intentional oversampling of specific subgroups. These weights can be integrated into statistical analyses, ensuring that weighted statistics accurately reflect the characteristics of the entire U.S. population [14]. This study was exempt from Institutional Review Board review because the study utilized de-identified, publicly available datasets.

### 2.2. Diagnoses of Solid Cancers

Participants were asked, “Have you ever been told by a doctor or other health professional that you had cancer or a malignancy of any kind?” Those who responded with an affirmative response were also asked, “What kind of cancer was it?” and up to three different cancers were recorded. Patients with a history of skin and hematological malignancies, such as leukemia, lymphoma, or myeloma, were excluded [15,16].

### 2.3. Ratio of Household Income to Federal Poverty Level

The household income-to-poverty ratio data were obtained from the NHIS, and are calculated as total household income divided by the family’s corresponding federal poverty level multiplied by 100. The FPL is issued annually by the Department of Health and Human Services and is determined by family size. The household income relative to the FPL is used to determine eligibility for federal programs and benefits, including Medicaid. Additional information regarding the income-to-poverty ratio data can be obtained from NHIS documentation [17]. For this analysis, household income was categorized as less than 200% FPL, 200% to 399% FPL, and 400% or greater FPL [18].

### 2.4. Outcomes

#### 2.4.1. Chronic Pain

The the primary outcome of this study was the presence of chronic pain. Information on pain was obtained through responses to the following questions: “In the past 3 months, how often did you have pain? Would you say never, some days, most days, or every day?” and “Over the past 3 months, how often did pain limit your life or work activities? Would you say never, some days, most days, or every day?” Chronic pain was defined as pain on most days or every day in the past 3 months vs. no pain on most days or every day in the past 3 months [6]. High-impact pain was assessed among the participants who had reported chronic pain. High-impact chronic pain (secondary outcome) was defined as pain that significantly restricts life or work activities on most days or every day vs. no pain that significantly restricts life or work activities on most days or every day [6].

#### 2.4.2. Pain Management Techniques Other than Over-the-Counter Medications

##### Opioids

Participants were asked about opioid use if they reported the use of prescription medication [12]. The participants’ use of prescription opioids was determined by inquiring whether they had taken any opioid pain relievers prescribed by a doctor, dentist, or other healthcare professional within the previous 12 months. As per the NHIS criteria, prescription opioid medications included hydrocodone, Vicodin, Norco, Lortab, oxycodone, OxyContin, Percocet, and Percodan. Participants were instructed not to include over-the-counter pain relievers such as aspirin, Tylenol, Advil, or Aleve. If in doubt, the participant was asked to tell the interviewer about the drug, and the interviewer would then determine whether it was an opioid or not. Those who reported usage were further queried regarding their opioid consumption within the past 3 months and also about acute use for short-term pain [12]. The variables were categorized as yes vs. no.

##### Alternative Pain Management Methods

Participants who had pain for at least some days were asked about other pain management techniques: (1) physical, occupational, or rehabilitative therapy (yes vs. no); (2) spinal manipulation or other forms of chiropractic care (yes vs. no); (3) talk therapies such as cognitive–behavioral therapy (yes vs. no); (4) yoga or tai chi (yes vs. no); (5) exercise for pain (yes vs. no); (6) massage for pain (yes vs. no); and (7) meditation, guided imagery, or other relaxation (yes vs. no) [12].

### 2.5. Covariates

To address potential confounding between household income and chronic pain, we included the following self-reported sociodemographic, behavioral, and clinical characteristics as covariates: age (categorized as 18–44, 45–64, or ≥65 years); sex (female or male); race/ethnicity (NH (non-Hispanic) White, NH Black, Hispanic, or NH Asian or other); marital status (married/living with a partner or not married); education (less than high school, high school or GED, some college, or more); covered by insurance (yes vs. no); urbanicity (large central metro, large fringe metro, medium and small metro, non-metropolitan); smoking (never, currently, or formerly); BMI (underweight, healthy weight, overweight, or obese); hypertension (yes vs. no); diabetes (yes vs. no); arthritis (yes vs. no); depression (yes vs. no); anxiety (yes vs. no); survey year (2019 or 2020); number of cancers (1 vs. >1), and type of cancer (breast, lung, colorectal, prostate, head and neck (mouth, tongue, lip, larynx, trachea, throat, or pharynx), melanoma, bladder, gynecologic (ovarian, uterine, or cervical), hepatobiliary (liver, hepatocellular, bile duct, or pancreatic), and digestive (stomach or esophageal)).

### 2.6. Statistical Analysis

Data were analyzed using SAS 9.4^®^ (SAS Institute, Inc.; Cary, NC, USA). All analyses utilized the appropriate NHIS sample weights, stratum, and primary sampling units to address oversampling, non-response, and non-coverage issues, ensuring nationally representative estimates were obtained. Sample characteristics are presented overall and by income-to-poverty level as number of participants and weighted percentages.

To explore the relationship between household income and chronic pain and high-impact pain, weighted logistic regression models were applied using PROC SURVEYLOGISTIC (https://support.sas.com/documentation/onlinedoc/stat/142/surveylogistic.pdf (accessed on 31 January 2024)) to estimate the odds ratio (OR) and corresponding 95% confidence intervals (CIs). Our dataset involved complex survey data with sampling weights, stratification, and clustering. To account for these design elements and ensure that our estimates are representative of the population, we selected weighted logistic regression models. This approach allows for the incorporation of survey weights, providing robust standard errors and accurate confidence intervals necessary for valid inferences about the association between household income and chronic pain. Model 1 adjusted for sex, age, education, race/ethnicity, marital status, BMI, smoking, insurance, and urbanicity. Model 2 additionally included adjustments for hypertension, diabetes, depression, anxiety, and arthritis. The >400% FPL group was designated as the reference level. Parallel analyses were conducted with high-impact pain as the outcome variable. Trend tests for FPL were performed by treating FPL as an ordinal variable within the multivariable logistic regression model. Two-sided *p* values < 0.05 were considered statistically significant in this study.

## 3. Results

### 3.1. Study Participants and Design

We conducted a cross-sectional analysis involving 4585 participants with a history of solid cancers, representing an estimated 17,052,286 U.S. adults aged 18 or older (Figure 1). Table 1 presents weighted descriptive statistics for sample characteristics across income categories. Of the analytic sample of participants, 1333 (29.6%) had household incomes of <200% FPL, 1372 (29.3%) had 200–399% FPL, and 1880 (41.1%) had ≥400% FPL. The majority of the participants were ≥65 years old (57.7%), female (58.9%), NH White (82.9%), married/living with a partner (65.2%), never smokers (51.6%), had some college or more (61.7%), and had hypertension (54.9%). Participants in the <200% FPL category tended to be female (66.1%), not married (53.2%), obese (39.1%), and exhibited higher comorbidities than higher-income groups. Missing data are reported in Appendix A.

### 3.2. Household Income and Chronic Pain

A total of 1649 participants (36.3%) with a history of cancer reported experiencing chronic pain. Among them, 502 participants (43.5%) reported high-impact chronic pain. Notably, the prevalence of chronic pain (Figure 2a) and high-impact chronic pain (Figure 2b) demonstrated a dose–response effect with a decreasing income-to-poverty ratio. We performed multivariable logistic regression to estimate the association between income-to-poverty ratio and chronic pain, adjusting for covariates (Table 2). In the adjusted Model 1, participants with household incomes < 200% FPL (aOR = 2.00, 95% CI: 159, 2.52) and 200–399% FPL (aOR = 1.38, 95% CI: 1.13, 1.69) had significantly higher odds of chronic pain compared to those with ≥400% FPL (Model 1). However, when additionally adjusting for comorbidities (Model 2), only <200% FPL remained significantly associated with chronic pain (aOR = 1.60, 95% CI: 1·25, 2·05). Among those with chronic pain, a household income <200% FPL was significantly associated with high-impact chronic pain compared to ≥400% FPL. Fully adjusted Model 2 results (Appendix A) reveal that chronic pain is significantly associated with age, obesity, arthritis, diabetes, depression, and anxiety, while high-impact pain is significantly associated with hypertension and anxiety.

### 3.3. Opioid Use

We assessed the prevalence of opioid and non-opioid pain management methods among cancer survivors (Table 3 and Appendix A). Across the 12-month, 3-month, and acute time frames, opioid usage was more common among individuals in the <200% FPL category, followed by the 200–399% category and >400% category, exhibiting a dose–response pattern (Table 3 and Appendix A).

### 3.4. Non-Opioid Pain Management

Thirty-one percent of participants relied solely on non-opioid management strategies to manage their pain. The most prevalent non-opioid pain management technique utilized by cancer survivors for chronic pain was “Other methods for pain” (excluding physical therapy, chiropractic care, talk therapy, cognitive–behavioral therapy, yoga, tai chi, qi gong, massage, and meditation), employed by 19.7% of participants. Among survivors experiencing high-impact chronic pain, 35% relied on “Other methods” for pain relief. However, the prevalence of these techniques varied across the income categories. Following “other” pain relief methods, cancer survivors in the ≥400% FPL category reported the use of physical therapy (12.5%), massage (11.9%), meditation (9.1%), chiropractic care (6.8%), yoga, tai chi, or qi gong (6.8%), and talk therapy (1.3%) (see Table 3). In contrast, cancer survivors in the <200% FPL category showed a preference for meditation (11.9%) and talk therapy (2.1%). The prevalence of pain-relieving methods for high-impact pain was comparable to that for chronic pain (Appendix A).

## 4. Discussion

In this cross-sectional study of a nationally representative sample of U.S. adults aged 18 years or older with a history of solid cancers, household income emerged as a significant social determinant influencing chronic pain. Participants with household incomes <200% FPL exhibited nearly twice the prevalence of chronic pain compared to those with ≥400% FPL. Our findings reveal a clear dose–response relationship between household income and chronic pain, with notably higher prevalence observed among individuals from lower-income households. This trend persisted when examining high-impact chronic pain as the outcome, underscoring the enduring disparities in pain experience linked to socioeconomic status.

### 4.1. Disparity in Cancer Pain

Our research aligns with extensive prior literature documenting the association between cancer and chronic pain [19,20,21]. However, to our knowledge, there are no prior nationally representative studies that have specifically evaluated the role of socioeconomic status in chronic pain and high-impact chronic pain among cancer survivors, as well as its association with pain management strategies. Chronic pain is best understood as a dynamic interaction of biological, psychological, and social factors [22], highlighting the intricate nature of its manifestation and management. Individuals in lower-SES groups encounter barriers to accessing healthcare services [23,24], including adequate pain management [25] and treatment options [24]. In this study, we found that 47.3% of adult cancer survivors with household income < 200% FPL experienced chronic pain, and 28.3% of them reported using opioids for pain management within the last 12 months. In contrast, only 19.0% of cancer survivors at >400% of the FPL used opioids. Similar differences were noted in the acute use of opioids, too. The differences in opioid use across income levels highlight complex factors influencing pain management choices [25,26,27]. The lower use of opioids at higher income levels can partly be explained by better access to healthcare information and resources. Higher-income individuals are more likely to be aware of newer guidelines aimed at reducing opioid prescriptions due to the significant rise in non-medical opioid use, the stigma associated with opioid addiction, and related problems [28]. This increased awareness and access to information can lead to a reduction in opioid use among higher-income individuals compared to lower-income groups [29,30]. Developing guidelines akin to the World Health Organization’s (WHO) three-step ladder for opioid use in individuals undergoing active cancer treatment could prove beneficial [31,32,33].

### 4.2. Socioeconomic Disparities in Cancer Pain Management

Financial constraints may limit access to non-opioid therapies or medical procedures that could alleviate chronic pain [34,35,36,37]. In our study, we found greater use of opioids in the lowest-income group, which suggests that recent practice guidelines that emphasize the use of non-opioid approaches may be unequally applied across socioeconomic statuses [31]. Notably, there were no substantial differences among the income levels for managing pain with physical therapy and meditation as the most common non-opioid management methods. These findings indicate the need for strategies to ensure safe and effective therapeutic approaches for chronic pain that are accessible to individuals across the entire socioeconomic spectrum.

Addressing the association between SES and chronic pain in cancer survivors requires comprehensive strategies aimed at reducing socioeconomic inequalities in access to healthcare services. Evidence suggests that the management of cancer pain has improved, but undertreatment remains common and impacts quality of life [38,39]. Policies aimed at expanding healthcare coverage, reducing out-of-pocket costs for essential medical services, and increasing funding for pain management programs in underserved communities are crucial steps in addressing this issue. Furthermore, initiatives aimed at enhancing outcomes in vulnerable cancer populations through education, employment opportunities, and community-based interventions can mitigate socioeconomic determinants of chronic pain and improve overall health outcomes. Our study has some notable strengths and limitations. This study is strengthened by using data from the NHIS, a large-scale survey of a nationally representative sample of the U.S. noninstitutionalized adult population [12]. The NHIS is a principal source for monitoring the health of the U.S. population. Additionally, the NHIS question about chronic pain was specifically designed based on the definition of pain by the International Association for the Study of Pain. Furthermore, by using the income-to-poverty ratio rather than solely an income measure, we provide a more thorough analysis of socioeconomic status, as it takes into account the household size.

### 4.3. Limitations

Our study has several limitations. The cancer diagnosis in the NHIS relies on self-reporting, which may result in some under-reporting. Some individuals may not be aware of their cancer and may not have been diagnosed yet, which is particularly relevant as individuals of lower SES tend to be diagnosed at more advanced cancer stages than their higher-SES counterparts [40]. Consequently, our analysis may have excluded individuals with undiagnosed cancers, potentially leading to an underestimation of the effect of lower income on chronic pain, although this may not have significantly impacted the results regarding lower-income households. We could not examine variations across different cancer types and cancer characteristics, such as treatment modalities or time elapsed since diagnosis, due to limitations of the available data. Our study primarily documented prevalent chronic pain and adjusted for common comorbidities associated with pain. However, a more detailed recognition of pain types could improve the interpretation of our findings and their implications for pain management in cancer care.

Additionally, the self-reported information in the NHIS data may limit the generalizability of our results to all populations. Despite NHIS weighting procedures designed to mitigate coverage and non-response bias, some measurement biases may still persist. Finally, a more comprehensive analysis of social determinants of health, such as food insecurity or lack of housing or transportation, may help identify potential targets for intervention.

## 5. Conclusions

Our study highlights the significant inverse relationship between household income and chronic pain prevalence, underscoring the need for targeted interventions to address socioeconomic disparities in chronic pain among cancer survivors. Additionally, longitudinal studies could provide insights into how changes in socioeconomic status impact chronic pain and quality of life over time. Addressing disparities is essential to improve pain management and overall quality of life for cancer survivors, particularly those from lower-income households.

## Figures and Tables

**Figure 1 cancers-16-02847-f001:**
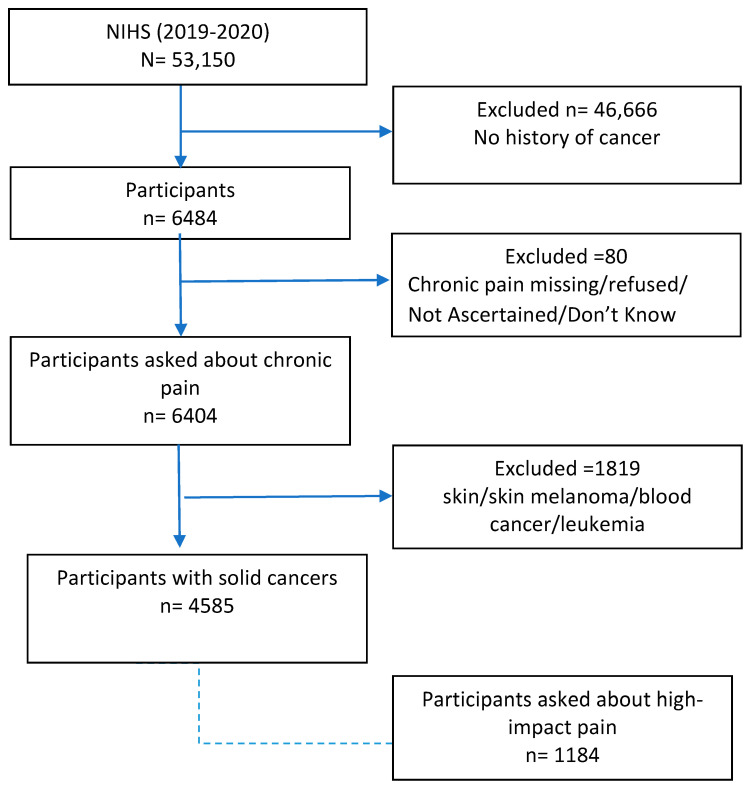
Participant flow chart.

**Figure 2 cancers-16-02847-f002:**
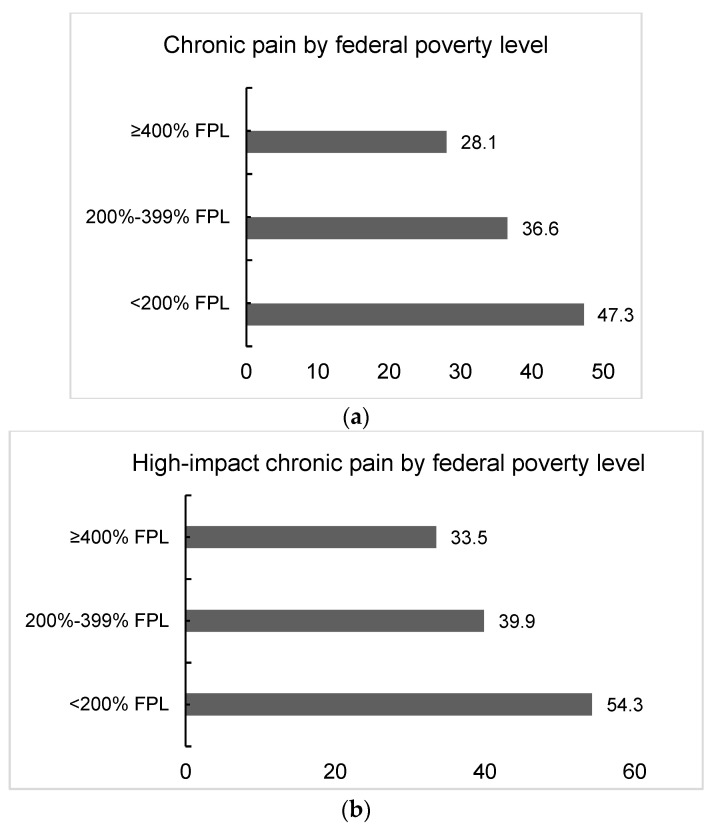
(**a**) Distribution of chronic pain by federal poverty level (*n* = 4585). (**b**) Distribution of high-impact pain with physical limitations by federal poverty level (*n* = 1184).

**Table 1 cancers-16-02847-t001:** Participant characteristics of the study population by federal poverty level, NHIS 2019–2020.

Characteristics	Overall		<200% FPL		200–399% FPL		≥400% FPL	
Unweighted *n*, weighted %	*n* = 4585 (%)		*n* = 1333 (%)		*n* = 1372 (%)		*n* = 1880 (%)	
**Age (years)**								
18–44	320	9.5	121	13.4	87	8.0	112	7.6
45–64	1285	32.7	362	30.2	317	27.6	606	38.2
≥65	2976	57.7	850	556.3	966	64.2	1160	54.1
**Sex**								
Female	2762	58.9	908	66.1	816	57.3	1038	54.9
**Race/ethnicity**								
Hispanic	280	7.0	141	14.9	76	8.0	63	4.3
NH White	3777	82.9	949	65.4	1157	80.8	1671	86.1
NH Black	354	6.4	167	13.4	98	8.5	89	5.1
NH Asian	76	1.8	17	1.9	21	1.4	38	3.4
Other	98	1.9	59	4.5	20	1.3	19	1.1
**Marital status**								
Not married	2212	33.8	899	53.2	688	36.3	625	22.6
Married/living with partner	2321	65.2	413	45.1	668	62.5	1240	76.5
**Education**								
Less than high school	354	9.5	243	23.8	86	9.7	25	1.9
High school or GED	1241	28.4	493	38.6	457	36.4	291	18.5
Some college or more	2963	61.7	583	36.5	825	53.6	1555	79.2
**Insurance**								
Covered	4449	96.0	1260	90.7	1334	96.4	1855	98.4
Not covered	132	3.9	72	9.2	36	3.4	24	1.5
**BMI**								
Underweight	66	1.4	21	1.5	27	2.3	18	1.0
Healthy weight	1390	29.0	346	24.3	409	29.0	635	31.8
Overweight	1613	35.7	452	33.2	471	34.5	690	35.9
Obese	1412	31.6	489	39.1	420	31.0	503	29.2
**Urbanicity**								
Large central metro	1089	23.8	305	24.9	284	27.5	500	27.5
Large fringe metro	1126	25.4	229	18.0	334	30.2	563	30.2
Medium and small metro	1515	32.3	461	31.4	476	30.4	578	30.4
Non-metropolitan	855	18.5	338	25.7	278	11.9	239	11.9
**Smoking history**								
Never	2287	51.6	573	45.7	662	46.4	1052	57.1
Current	524	11.8	267	21.0	158	14.0	99	6.2
Former	1736	35.9	475	31.7	540	38.8	721	36.3
**Comorbidities**								
Diabetes (Yes)	787	17.8	311	23.1	258	20.7	218	11.9
Hypertension (Yes)	2586	54.9	876	63.6	814	58.3	896	46.2
Arthritis (Yes)	2136	44.0	725	51.4	675	46.7	736	36.6
Depression (Yes)	987	21.4	425	29.4	275	20.6	287	16.4
Anxiety (Yes)	805	17.6	344	24.9	219	16.5	242	13.2
**Number of cancers (>1)**	1028	20.8	287	19.7	301	20.9	440	21.4
**Cancer type #**								
Colorectal	364	7.6	133	10.3	116	7.4	115	5.7
Breast (females)	1236	42.0	369	37.6	367	42.9	500	45.2
Lung	188	4.0	85	6.0	54	3.9	49	2.7
Prostate (males)	806	43.0	153	34.5	243	42.9	410	45.5
Head and neck	89	1.9	27	1.9	28	2.3	34	1.6
Melanoma	365	7.7	68	4.8	113	8.1	184	9.6
Bladder	177	3.5	42	2.6	65	4.5	70	3.5
Gynecologic (females)	646	26.1	276	33.7	180	24.4	190	20.7
Hepatobiliary	80	1.7	23	1.6	27	2.1	30	1.6
Digestive	60	1.4	25	2.0	14	1.1	21	1.2

Column percentages may not aggregate to a hundred percent; # percentages represent the self-reported type of cancer; participant could have more than one cancer. Abbreviations: FPL—federal poverty level; NH—non-Hispanic; U.S.—United States; GED—General Education Diploma; AIAN—American Indian/Alaska Native; BMI—body mass index.

**Table 2 cancers-16-02847-t002:** Association between federal poverty level and chronic pain.

	*n*/N	Weighted %	aOR and 95% CI *	aOR and 95% CI †
**Chronic pain (*n* = 4585)**			**Model 1**	**Model 2**
<200% FPL	650/1333	47.3	2.00 (1.59–2.52)	1.60 (1.25–2.05)
200–399% FPL	481/1372	36.6	1.38 (1.13–1.69)	1.18 (0.96–1.46)
≥400% FPL	518/1880	28.1	REF	REF
			*p*-trend < 0.0028	*p*-trend < 0.0005
**High-impact chronic pain interferes with life and work; physical limitations (*n* = 1184)**				
<200% FPL	256/476	54.3	1.86 (1.17–2.94)	1.73 (1.09–2.74)
200–399% FPL	129/338	39.9	1.34 (0.88–2.04)	1.24 (0.80–1.90)
≥400% FPL	117/370	33.5	REF	REF
			*p*-trend = 0.0082	*p*-trend = 0.0203

Abbreviations: aOR—adjusted odds ratio. * The model is adjusted for age, sex, race/ethnicity, education, marital status, smoking, insurance coverage, BMI, and urban/rural status. † The model is adjusted for age, sex, race/ethnicity, education, marital status, smoking, insurance coverage, BMI, urban/rural status, diabetes, hypertension, depression, anxiety, and arthritis.

**Table 3 cancers-16-02847-t003:** Prevalence of chronic pain-relieving methods adopted by cancer survivors by federal poverty level.

Variables	Overall *n* = 4585	<200% FPL *n* = 1333 (%)	200–399% FPL *n* = 1372 (%)	≥400% FPL *n* = 1880 (%)
Opioid use				
Opioid use in the past 12 months				
Yes	1002 (22.4)	383 (28.3)	271 (21.3)	348 (19.0)
No	3183 (67.0)	862 (62.6)	987 (69.0)	1334 (68.7)
Not asked/refused/missing	400 (10.6)	88 (9.1)	114 (9.7)	198 (12.3)
Opioid use in the past 3 months				
Yes	620 (13.7)	260 (18.9)	175 (13.8)	185 (9.8)
No	380 (8.7)	123 (9.4)	95 (7.3)	162 (9.1)
Not asked/refused/missing	3585 (77.6)	950 (71.7)	1102 (78.8)	1533 (81.1)
Acute opioid use in the past 3 months				
Yes	383 (8.2)	155 (10.4)	108 (8.5)	120 (6.5)
No	237 (5.4)	105 (8.5)	17 (5.3)	15 (3.4)
Not asked/refused/missing	3965 (86.3)	1073 (81.1)	1197 (86.2)	1695 (90.2)
Non-opioid methods				
Physical therapy				
Yes	557 (11.6)	164 (11.5)	160 (10.5)	233 (12.5)
No	2902 (64.1)	943 (70.4)	864 (65.3)	1095 (58.5)
Not asked/refused/missing	1126 (24.3)	226 (18.1)	348 (24.1)	552 (29.0)
Chiropractic care				
Yes	289 (6.2)	70 (5.4)	89 (6.3)	130 (6.8)
No	3168 (69.4)	1035 (76.3)	936 (69.6)	1197 (64.2)
Not asked/refused/missing	1128 (24.4)	228 (18.3)	347 (24.1)	553 (29.0)
Talk, cognitive–behavioral therapy				
Yes	81 (1.7)	31 (2.1)	25 (2.0)	25 (1.3)
No	3375 (73.9)	1074 (79.6)	998 (73.8)	1303 (69.8)
Not asked/refused/missing	1129 (24.4)	228 (18.3)	349 (24.2)	552 (30.0)
Yoga, tai chi, or qi gong				
Yes	279 (5.5)	72 (5.1)	64 (4.1)	143 (6.8)
No	3179 (70.1)	1034 (76.6)	960 (71.8)	1185 (64.3)
Not asked/refused/missing	1127 (24.4)	227 (18.2)	348 (24.1)	552 (30.0)
Massage				
Yes	446 (9.9)	99 (7.6)	127 (9.3)	220 (11.9)
No	3011 (65.7)	1006 (74.1)	897 (66.6)	1108 (59.1)
Not asked/refused/missing	1128 (24.4)	228 (18.3)	348 (24.1)	552 (29.0)
Meditation				
Yes	483 (10.0)	162 (11.9)	140 (9.6)	181 (9.1)
No	2973 (65.5)	943 (69.8)	883 (66.2)	1147 (61.9)
Not asked/refused/missing	1129 (24.4)	228 (18.3)	349 (24.2)	552 (29.0)
Other				
Yes	961 (19.7)	311 (20.7)	271 (19.5)	379 (19.2)
No	2496 (55.9)	794 (61.1)	753 (56.4)	949 (51.9)
Not asked/refused/missing	1128 (24.4)	228 (18.2)	348 (24.1)	552 (29.0)

Abbreviations: FPL—federal poverty level.

## Data Availability

Data utilized for this study are available at the following link: https://www.cdc.gov/nchs/nhis/data-questionnaires-documentation.htm (accessed on 31 January 2024).

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
