# Peer review of "Household Income Is Associated with Chronic Pain and High-Impact Chronic Pain among Cancer Survivors: A Cross-Sectional Study Using NHIS Data"

_cancers, 2024, doi:10.3390/cancers16162847_

Round 1

Reviewer 1 Report

Comments and Suggestions for Authors

The current study aims to evaluate the correlation between chronic pain and family household income levels in cancer survivors. However, the study's patient cohort is too heterogeneous. Pain is a prevalent issue among cancer survivors and can be due to various causes not related to cancer, such as trigeminal neuralgia, migraine headaches, back pain not related to spinal metastasis, or comorbidities like multiple sclerosis. In the current manuscript, it only documents pain in cancer patients, which does not align with my understanding of discussing pain due to cancer. Furthermore, the diagnosis of solid cancers in the manuscript is vague. Skin cancer was excluded, but melanoma was included. What about patients with brain metastases? Additionally, the number of lung cancer patients is exceedingly low, accounting for only 2.7% of the whole cohort. This does not reflect the general population. I believe the authors should address these points or at least document them in the limitations.

Author Response

  • The current study aims to evaluate the correlation between chronic pain and family household income levels in cancer survivors. However, the study's patient cohort is too heterogeneous.

Response: We appreciate the reviewer's feedback. We acknowledge the heterogeneity of the cohort, which stems from the nature of the NHIS, a nationally representative cross-sectional household survey of the U.S. population. Participants in NHIS are selected to monitor the health of the US population. This heterogeneity is a strength, as the NHIS uniquely provides comprehensive information on both health and socioeconomic characteristics, allowing us to explore the impact of annual household income on chronic pain across various cancer types. By selecting only cancer cohort, we are able to investigate the impact of annual household income on chronic pain within a diverse group of cancer survivors.

  • Pain is a prevalent issue among cancer survivors and can be due to various causes not related to cancer, such as trigeminal neuralgia, migraine headaches, back pain not related to spinal metastasis, or comorbidities like multiple sclerosis. In the current manuscript, it only documents pain in cancer patients, which does not align with my understanding of discussing pain due to cancer.

Response: We appreciate the reviewer's comment and understand the concern regarding the documentation of pain sources. In our manuscript, we aimed to document the prevalence of pain among cancer survivors, recognizing that pain in this population can stem from various sources, including but not limited to cancer itself. This comprehensive approach allows us to capture the overall burden of pain experienced by cancer survivors. We acknowledge that distinguishing between cancer-related pain and pain due to other causes is important and will clarify this distinction in the manuscript as a limitation to better align with the reviewer's understanding of discussing pain specifically related to cancer.

  • Furthermore, the diagnosis of solid cancers in the manuscript is vague. Skin cancer was excluded, but melanoma was included. What about patients with brain metastases?

Response: We appreciate the reviewer's observation. Our definitions of solid cancers, including the inclusion of melanoma, are based on prior published studies by Jones et al. (2024) and Haenen et al. (2022), which also characterized pain in cancer survivors. We have now referenced the articles for definition of solid cancers. Patients with brain metastases were not specifically excluded because metastasis information is not available in NHIS.

  • Additionally, the number of lung cancer patients is exceedingly low, accounting for only 2.7% of the whole cohort. This does not reflect the general population. I believe the authors should address these points or at least document them in the limitations.

Response: We appreciate the reviewer’s observation. We would like to clarify that lung cancer represents 4.0% of our study cohort, rather than the 2.7% mentioned. However, we acknowledge that this proportion is lower compared to the 12.0% of lung cancer cases diagnosed nationally in 2024 (https://seer.cancer.gov/statfacts/html/common.html). We have included the following sentence on pg. 13 and line 327-329 “Additionally, the self-reported information in the NHIS data may limit the generalizability of our results to all populations. Despite NHIS weighting procedures designed to mitigate coverage and non-response bias, some measurement biases may still persist.”

Reviewer 2 Report

Comments and Suggestions for Authors

The authors evaluated whether household income is associated with chronic pain and high- impact chronic pain among cancer survivors using database. I think the study is interesting. I have, however, several concerns. First, the manuscript does not follow STROBE guidelines (PMID: 17938396). Please follow the guidelines. Second, it is not clear why the authors select the models. Third, I do not see odds ratio for the variables (sex, age, …).

Major comments

#1. The manuscript does not follow STROBE guidelines (PMID: 17938396). Please follow the guidelines.

#2. It is not clear why the authors select the models (Page 4, Line 150). Please show the reasons.

#3. I do not see odds ratios for the variables (sex, age, …) (Page 4, Line 150/ Table 3). Please show the odds ratios.

Comments on the Quality of English Language

None.

Author Response

Please see attached document with our response. 

Reviewer 3 Report

Comments and Suggestions for Authors

Interesting manuscript. the authors capture a significant issue for individuals with cancer in particular survivors experiencing pain.

The concept is still in need of investigation. You did focus on the SES which is certainly, relevant in USA (and other countries) where each individual does not have access to comprehensive health care.

I do have issues with the origin of data which is a major limitation in line with some questions I added inside the manuscript.

The conclusion should include more directions for future studies  because of the importance  of the preliminary information the study entailed.

Author Response

Please see attached response to the comments

Round 2

Reviewer 1 Report

Comments and Suggestions for Authors

I am satisfied with the revision. 

Reviewer 2 Report

Comments and Suggestions for Authors

The authors improved the manuscript. I have no further comments.